# Impact of pole dancing on mental wellbeing and sexual self-concept: Protocol for a systematic review and meta-analysis

Xizi Li[1], Jianyu Shen[2], Kai Cui[3], Ying Wang[1]*

1 School of Dance and Martial Arts, Capital University of Physical Education and Sports, Haidian, Beijing, China, 2 Department of Physical Education, Engineering & Technical College of Chengdu University of Technology, Leshan, Chengdu, China, 3 Beijing Xiehe Hospital, Chaoyang, Beijing, China

* 302447720@qq.com

**Data Availability Statement:** No datasets were generated or analysed during the current study. All relevant data from this study will be made available upon study completion.

## Abstract

### Background

Despite the recognized psychological benefits of traditional dance forms, the impact of newer forms, such as pole dancing, on mental well-being and sexual self-concept remains underexplored. This protocol outlines a systematic review and meta-analysis aimed at elucidating the effects of pole dancing, a burgeoning non-pharmacological intervention, on these dimensions of mental health.

### Methods

This systematic review was registered in the PROSPERO. We will follow the Preferred Reporting Items for Systematic Reviews and Meta-analysis Protocol to accomplish the systematic review protocol. This review will systematically search electronic databases, including PubMed, Embase, Web of Science, Medline, and CNKI, for randomized controlled trials (RCTs) assessing the impact of pole dancing on mental well-being and sexual self-concept. Two independent evaluators will screen the literature, extract data, and evaluate study quality and bias. Data synthesis will utilize Stata 14.0 and Revman 5.4, employing random-effects models. The Grading of Recommendations, Development, and Evaluation (GRADE) system will appraise evidence reliability, with subgroup analysis exploring heterogeneity sources. Publication bias will be assessed through funnel plots and Egger's regression tests.

### Discussion

This review aims to fill the gap in the current literature by providing a comprehensive evaluation of pole dancing's psychological effects. It is anticipated that this systematic review and meta-analysis will offer valuable insights for health policy and practice, advocating for the inclusion of pole dancing in mental health and sexual well-being interventions.

**Funding:** The author(s) received no specific funding for this work.

**Competing interests:** The authors have declared that no competing interests exist.

## Trial registration

**Systematic review registration:** PROSPERO CRD42024529369.

## Introduction

The influence of dance extends far beyond mere physical health, permeating various facets of psychological well-being. Evidence robustly suggests that dance not only bolsters mental functioning, self-confidence, and self-esteem but also diminishes feelings of isolation [1, 2]. Furthermore, it positively affects stress levels, psychological capital, and self-concept, particularly noted among students with artistic inclinations [3]. The contribution of dance to psychological well-being is manifold, evidenced by reductions in anxiety, boosts in self-esteem, and mood enhancement [4]. Its beneficial impacts are universally acknowledged, aiding emotional and physical health across diverse groups, including individuals with intellectual disabilities, older adults, and migrant pupils [5, 6].

While extant research on dance therapy has predominantly concentrated on traditional and expressionistic dance forms, exploring contemporary dance genres, particularly those with distinct techniques, remains sparse. Among these, pole dancing emerges as a compelling subject of study. Initially perceived as an activity confined to strip clubs, pole dancing has transcended its origins to be recognized as both a reputable form of fitness and an expressive art form. This evolution is reflected in the practice's demands for considerable strength, flexibility, and coordination, enhancing physical fitness and promoting profound body awareness [7].

Emerging research underscores the potential of pole dancing to foster significant psychological benefits, including enhanced self-confidence, self-esteem, the fostering of relationships, and a reinforced sense of belonging—each a cornerstone of resilience and mental wellbeing [8, 9]. Specifically, pole dancing has been linked to reductions in depression and anxiety, increases in positive affect, self-esteem, and self-efficacy, and reductions in negative affect and loneliness [10, 11]. Furthermore, it challenges entrenched gender norms, offering a space for the exploration and redefinition of masculinity and sexuality, particularly noted in the participation of men in pole dancing activities, which may counter traditional hetero masculine norms [12].

The intricate relationship between sexual self-concept and mental health has been explored across various contexts, including among individuals with physical-motor disabilities and within the dynamics of marital burnout, suggesting the integral role of sexual self-concept in overall well-being and interpersonal relations [13, 14]. Pole dancing empowers participants to embrace their bodies, express individuality through movement, and challenge societal norms and stereotypes, fostering self-acceptance and body celebration [15, 16]. It offers a unique lens through which the implications of sexual self-concept discrepancies on mental health can be examined, especially among young black women [17].

Pole dancing has been shown to cultivate positive body image experiences by developing physical skills that promote body acceptance, self-confidence, personal growth, and body appreciation, culminating in empowerment and a positive self-image [18]. The exploration of pole dancing's impact on mental well-being and sexual self-concept reveals its capacity to positively influence psychosexual development and overall mental health [19]. Despite the availability of some studies on the subject, a systematic review and meta-analysis consolidating the effects of pole dancing on mental well-being and sexual self-concept remains absent. This review aims to fill that gap, considering how age, gender ratio, pole dancing style, participant

dropout rates, measurement tools, and geographical context may affect pole dancing's efficacy. The findings of this systematic review and meta-analysis hold the promise of offering significant insights into improving mental well-being and addressing issues related to sexual self-concept.

## Methods

### Protocol and registration

This study is registered with the International Prospective Register of Systematic Reviews, PROSPERO, under CRD42024529369. To ensure the rigor and transparency of our systematic review, we adhere to the Preferred Reporting Items for Systematic Reviews and Meta-Analyses Protocols (PRISMA-P) guidelines [20]. Given that our research entails synthesizing existing primary data rather than collecting new primary data, it does not necessitate ethical approval. This approach underscores our commitment to conducting a systematic review that meets the highest methodological integrity and ethical consideration standards.

### Data sources and search strategies

To comprehensively capture the scope of research on pole dancing's impact on mental well-being and sexual self-concept, a meticulous literature search will be conducted across five electronic databases: Web of Science, Embase, PubMed, Medline, and the China National Knowledge Infrastructure (CNKI). The search will span the entirety of the available records in each database up to March 2024, ensuring a wide and contemporary range of literature is reviewed. Team members XL and KC have meticulously designed the search strategies to encompass all relevant terms and their variations in English: "mental well-being", "mental health", "mood states", "sexual self-efficacy", "sexual anxiety", "sexual motivation", "sexual consciousness", "self-esteem", "body appreciation", "body image appreciation", "positive body image", and "pole danc*". These terms are adjusted for equivalent searches within Chinese-language databases to ensure comprehensive global coverage. Please refer to Table 1 for details. The compilation of search results will be managed using Endnote X9 software, with a systematic approach for identifying and removing duplicates based on authorship, title, and publication date details to refine the final literature for review.

### Eligibility criteria

**Type of study.**   This systematic review and meta-analysis will rigorously focus on randomized controlled trials (RCTs), providing a robust methodological framework to assess the impact of pole dancing on mental well-being and sexual self-concept.

**Types of participants.**   We impose no restrictions on participant demographics, allowing for a broad inclusivity of populations in the analysis.

**Interventions and comparators.**   The comparative analysis will distinguish between control groups—participants engaging in daily activities—and experimental groups receiving specific pole dancing interventions.

**Types of outcome measures.**   Our primary outcomes of interest are the degrees of mental well-being, and our secondary outcome of interest is the enhancement or alteration of the sexual self-concept. In terms of mental wellbeing, We included studies using scales including the Warwick-Edinburgh Mental Well-being Scale (WEMWBS) and Ryff Psychological Well-being scale (PWBS-42). For enhancement and alteration of the sexual self-concept, We included studies using scales such as the Multidimensional Sexual Self-Concept Questionnaire (MSSCQ).

**Table 1. Search strategy.**

| Databases | Search strategies |
|---|---|
| Web of Science | #1: (((((((((((TS = ("mental wellbeing")) OR TS = ("mental health")) OR TS = ("mood states")) OR TS = ("sexual self-efficacy")) OR TS = ("sexual anxiety")) OR TS = ("sexual motivation")) OR TS = ("sexual consciousness")) OR TS = ("self-esteem")) OR TS = ("body appreciation")) OR TS = ("body image appreciation")) OR TS = ("positive body image") <br> #2: TS = ("pole danc*") <br> #3: #1 AND #2 |
| Embase | #1: 'mental wellbeing':ti,ab,kw OR 'mental health':ti,ab,kw OR 'mood states':ti,ab,kw OR 'sexual self-efficacy':ti,ab,kw OR 'sexual anxiety':ti,ab,kw OR 'sexual motivation':ti,ab,kw OR 'sexual consciousness':ti,ab,kw OR 'self esteem':ti,ab,kw OR 'body appreciation':ti,ab,kw OR 'body image appreciation':ti,ab,kw OR 'positive body image':ti,ab,kw <br> #2: 'pole danc*':ti,ab,kw <br> #3: #1 AND #2 |
| Medline | #1: AB "mental wellbeing" OR AB "mental health" OR AB "mood states" OR AB "sexual self-efficacy" OR AB "sexual anxiety" OR AB "sexual motivation" OR AB "sexual consciousness" OR AB "self-esteem" OR AB "body appreciation" OR AB "body image appreciation" OR AB "positive body image" <br> #2: AB "pole dance*" <br> #3: #1 AND #2 |
| Pubmed | #1: ((((((((((("mental wellbeing"[Title/Abstract]) OR ("mental health"[Title/Abstract])) OR ("mood states"[Title/Abstract])) OR ("sexual self-efficacy"[Title/Abstract])) OR ("sexual anxiety"[Title/Abstract])) OR ("sexual motivation"[Title/Abstract])) OR ("sexual consciousness"[Title/Abstract])) OR ("self-esteem"[Title/Abstract])) OR ("body appreciation"[Title/Abstract])) OR ("body image appreciation"[Title/Abstract])) OR ("positive body image"[Title/Abstract]) <br> #2: "pole dance*"[Title/Abstract] <br> #3: #1 AND #2 |
| CNKI | #1: SU = '性' or SU = '健康' or SU = '心理' or SU = '情绪' or SU = '心态' <br> #2: SU = '钢管舞' <br> #3: #1 and #2 |

**Language restriction.** Inclusivity extends to the linguistic and publication date criteria, where no restrictions will be applied, facilitating a comprehensive and diverse collection of research findings for analysis.

## Study selection and data extraction

The study selection and data extraction process will be meticulously undertaken by two independent reviewers, XL and KC, to ensure the integrity and accuracy of the systematic review. Initially, both reviewers will conduct a preliminary evaluation of all retrieved articles by carefully examining their titles, abstracts, and full texts. Predefined inclusion and exclusion criteria will guide this evaluation to determine each study's eligibility for inclusion in the review (Fig 1). In instances of disagreement or discrepancy between the two reviewers, a consensus will be sought through discussion. A third reviewer, YW, will be consulted to make a final decision if a resolution cannot be achieved.

Upon determining an article's eligibility, detailed data extraction will commence. This phase involves the meticulous collection of key study characteristics and findings. Specifically, the following information will be extracted from each eligible study: author's names, year of publication, country of origin, specific type of pole dancing intervention, total number of participants, gender distribution among participants, dropout rates, participants' age range, duration of the pole dancing intervention, measurement instruments used, and the average and standard deviation of continuous outcome measures. This comprehensive data extraction is essential for ensuring that the review captures a holistic understanding of the impact of pole dancing on mental well-being and sexual self-concept, facilitating a nuanced analysis and interpretation of the findings.

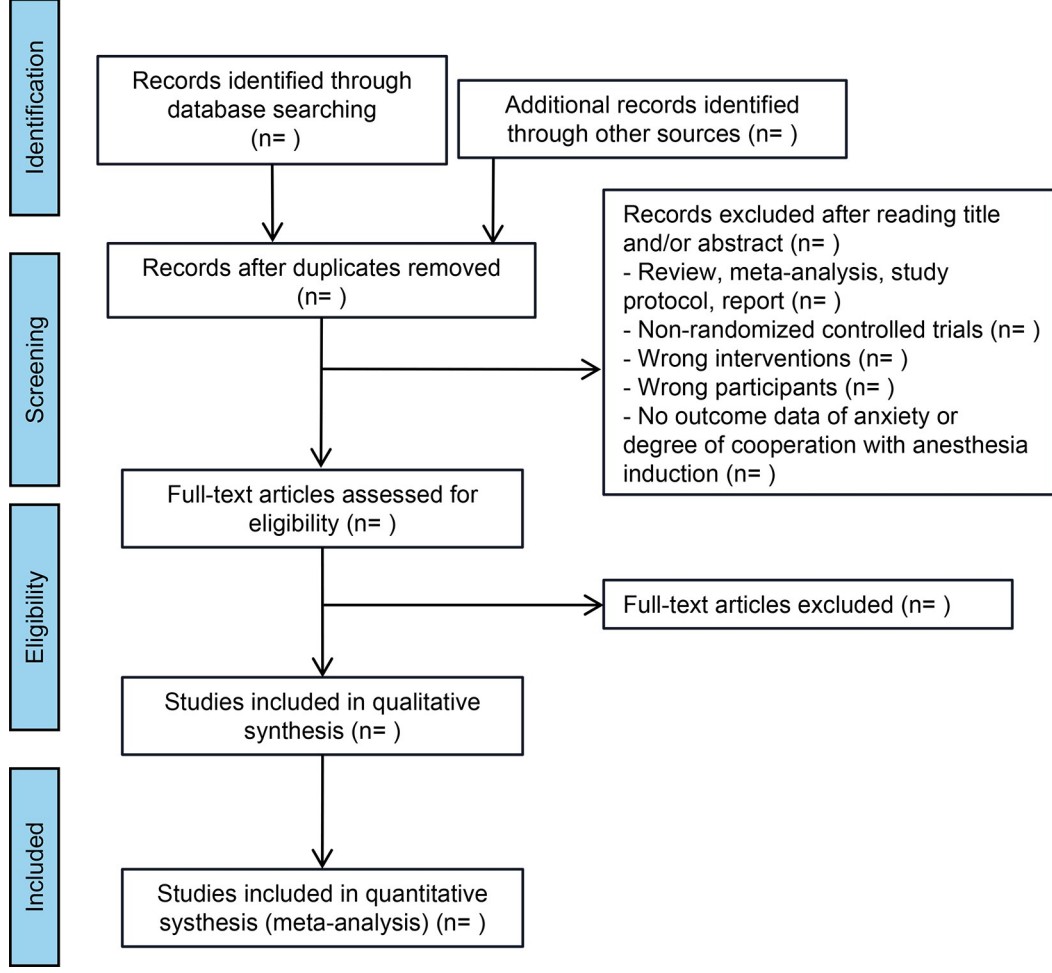

**Fig 1. Flowchart steps of the systematic review.**

## Methodological quality assessment

To ensure the integrity and validity of our systematic review, the methodological quality of each selected study will undergo a rigorous assessment. This assessment will be conducted by two independent researchers, XL and KC, utilizing the widely recognized Cochrane Collaboration tool for assessing the risk of bias in randomized controlled trials (RCTs) [21]. This tool facilitates a structured evaluation across several domains crucial to the credibility of study findings, categorized into three distinct levels of bias risk: low, unclear, and high.

The specific criteria evaluated will include random sequence generation to assess selection bias, allocation concealment to further evaluate selection bias, blinding of participants and personnel to address performance bias, analysis of incomplete outcome data to determine attrition bias, examination of selective reporting to identify reporting bias, and the identification of any other potential sources of bias. These evaluations will form the basis of a comprehensive bias risk profile for each study, which will be meticulously documented and analyzed.

The assessment process will leverage Review Manager 5.4 software for the graphical representation and detailed evaluation of bias risks. This approach not only ensures a transparent and systematic review of each study's methodological quality but also facilitates the

identification of studies with high levels of rigor and reliability. In cases where discrepancies arise between the two primary researchers, efforts will be made to resolve such differences through discussion. If a consensus cannot be reached, a third reviewer (KC) will be consulted to make a final determination. This multi-tiered review process underscores our commitment to upholding the highest standards of methodological scrutiny and ensuring the robustness of our systematic review's conclusions.

## Grading of evidence

The quality of evidence for each specific outcome will be classified according to the Grading of Recommendations Assessment, Development, and Evaluation (GRADE) approach [22]. This method stratifies evidence quality into four levels: high, moderate, low, and very low, providing a transparent and structured process for rating the certainty of evidence in systematic reviews.

## Statistical analysis

**Addressing missing data.** In instances of missing or inadequate data within articles, attempts will be made to contact the original authors for additional information. Should these efforts prove unsuccessful or the data remains insufficient, the affected studies will be excluded from the analysis.

**Analyzing treatment effects.** The meta-analysis will be conducted using Revman 5.4, developed by the Cochrane Collaboration. For continuous outcomes measured across studies using different scales, the standard mean difference will be calculated, weighted by the inverse of the variance, and presented with 95% confidence intervals (CIs) in forest plots. Statistical significance is predetermined at p-values less than 0.05. Heterogeneity among studies will be quantified using the $I^2$ statistic and the Chi-square test, with significant heterogeneity indicated by an $I^2$ value over 50% and a p-value under 0.10. Depending on the $I^2$ value, analyses will proceed under either a fixed or random-effects model. Subgroup analyses will be performed to explore potential sources of heterogeneity.

**Sensitivity analysis.** The robustness of the meta-analysis findings will be tested through sensitivity analysis, employing a one-by-one elimination method to pinpoint the sources of heterogeneity.

**Subgroup analysis.** To understand the effects of diverse variables on the outcomes, subgroup analyses will be carried out based on factors like age, gender ratio, type of pole dancing, dropout rates, measurement tools, and country of origin.

**Evaluating publication bias.** The publication bias will be assessed through funnel plot analysis for studies exceeding ten in number, complemented by Egger's regression test for statistical evaluation of bias.

If quantitative synthesis is deemed unsuitable, we will summarize and discuss the outcomes of each study, factoring in the potential for bias and the importance of the findings. After integrating the results, we will pinpoint those interventions that exhibit effectiveness and provide valuable insights for future research endeavors and informed decision-making.

## Ethics and dissemination

Given the study's reliance on published data rather than individual patient information, ethics approval is not required. The results of this systematic review and meta-analysis will be disseminated through presentations at relevant scientific conferences and publication in a peer-reviewed journal.

### Amendments

Any changes made to this protocol during the review process will be documented and disclosed in the final report to ensure transparency and accountability.

## Discussion

The burgeoning interest in pole dancing as a recreational and fitness activity brings to light its potential therapeutic effects on mental well-being and the sexual self-concept. Empirical evidence suggests that engaging in pole dancing can positively influence mental health parameters, including enhancements in sexual self-efficacy, reductions in sexual anxiety, improvements in sexual self-esteem, and an appreciation for one's body [20]. Despite the anecdotal and preliminary research findings supporting these benefits, the academic landscape lacks a comprehensive systematic review and meta-analysis dedicated to consolidating the evidence on the psychological impacts of pole dancing.

Our proposed systematic review and meta-analysis aim to fill this gap in the literature, offering a rigorous examination of the effects of pole dancing on mental well-being and sexual self-concept. Acknowledging that the impact of pole dancing may vary across different demographics and contexts, our analysis will consider variables such as age, gender ratio, the specific style of pole dancing, participant dropout rates, the measurement tools used, and the country of origin. Such a nuanced approach ensures that our findings will provide meaningful insights relevant to diverse individuals and settings.

By systematically collating and analyzing the existing body of research, this study is poised to be a seminal work in the field, establishing a foundation for future research, clinical practices, and policy decisions. It is anticipated that our findings will not only affirm the therapeutic value of pole dancing but also highlight its potential as a viable intervention for enhancing mental well-being and sexual self-concept. In doing so, this review will contribute significantly to understanding non-traditional physical activities as therapeutic tools, paving the way for their integration into holistic health and wellness programs. Through this comprehensive evaluation, we aim to deliver an up-to-date synthesis of evidence that underscores the unique benefits of pole dancing, thereby informing practitioners, policymakers, and participants about its efficacy and safety in promoting mental health and wellbeing.

This study, while comprehensive, is not without its limitations. A primary constraint is the exclusive focus on published studies, thereby excluding grey literature and unpublished works that could potentially offer valuable insights into the effects of pole dancing on mental well-being and sexual self-concept. While practical in terms of accessibility and verifiability of data, this decision may introduce publication bias, as studies with positive outcomes are more likely to be published than those with negative or inconclusive results.

Furthermore, the current landscape of randomized controlled trials (RCTs) specifically examining pole dancing's impact on mental well-being and sexual self-concept is relatively sparse, both in number and in methodological rigor. The existing studies, although informative, may not provide a sufficiently robust evidence base to draw definitive conclusions. This limitation underscores the need for further research in high-quality RCTs, which are essential to strengthen and substantiate the clinical evidence regarding pole dancing's therapeutic potential. Advancing this line of inquiry will require concerted efforts from the research community to design and implement studies that can effectively address these gaps and contribute to a more comprehensive understanding of pole dancing as a modality for enhancing mental health and well-being.

## Supporting information

**S1 Checklist. PRISMA-P 2020 checklist.**
(DOCX)

## Author Contributions

**Conceptualization:** Xizi Li, Jianyu Shen.

**Data curation:** Xizi Li.

**Formal analysis:** Xizi Li.

**Funding acquisition:** Xizi Li.

**Investigation:** Jianyu Shen.

**Methodology:** Jianyu Shen.

**Project administration:** Jianyu Shen.

**Resources:** Kai Cui.

**Software:** Kai Cui, Ying Wang.

**Supervision:** Ying Wang.

**Validation:** Ying Wang.

**Visualization:** Ying Wang.

**Writing – original draft:** Xizi Li, Kai Cui.

**Writing – review & editing:** Xizi Li, Kai Cui.

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
