## [Decision Letter · Decision Letter 0]

17 May 2024

PONE-D-24-13949

Impact of pole dancing on mental wellbeing and sexual self-concept: protocol for a systematic review and meta-analysis

PLOS ONE

Dear Dr. Li,

Thank you for submitting your manuscript to PLOS ONE. After careful consideration, we feel that it has merit but does not fully meet PLOS ONE’s publication criteria as it currently stands. Therefore, we invite you to submit a revised version of the manuscript that addresses the points raised during the review process.

We look forward to receiving your revised manuscript.

Kind regards,

Adetayo Olorunlana, Ph.D.

Academic Editor

PLOS ONE

Journal Requirements:

2. We note you have included a table to which you do not refer in the text of your manuscript. Please ensure that you refer to Table 1 in your text; if accepted, production will need this reference to link the reader to the Table.

3. We note that this manuscript is a systematic review or meta-analysis; our author guidelines therefore require that you use PRISMA guidance to help improve reporting quality of this type of study. Please upload copies of the completed PRISMA checklist as Supporting Information with a file name “PRISMA checklist”.

Reviewers' comments:

Reviewer's Responses to Questions

**Comments to the Author**

1. Does the manuscript provide a valid rationale for the proposed study, with clearly identified and justified research questions?

Reviewer #1: Yes

2. Is the protocol technically sound and planned in a manner that will lead to a meaningful outcome and allow testing the stated hypotheses?

Reviewer #1: Yes

3. Is the methodology feasible and described in sufficient detail to allow the work to be replicable?

Reviewer #1: Yes

4. Have the authors described where all data underlying the findings will be made available when the study is complete?

Reviewer #1: Yes

5. Is the manuscript presented in an intelligible fashion and written in standard English?

Reviewer #1: Yes

6. Review Comments to the Author

You may also provide optional suggestions and comments to authors that they might find helpful in planning their study.

Reviewer #1: The authors present a well-written systematic review protocol, with the aim of evaluating the effect of pole dancing on mental well-being and sexual self-concept. A relevant topic, with no published systematic reviews and with the potential to provide good evidence for decision-making.

However, I suggest minor revisions so that the manuscript is even better for readers.

The introduction is long and tiring. The suggestion for the authors is to summarize the content featuring the main theme, pole dancing, followed by text with the most up-to-date references of studies that show potential psychological benefits, maintaining the important gap of there being no systematic reviews specifically on pole dancing and concluding the introduction with the main objective.

Finally, the methodology is well constructed, but the text "Our primary outcomes of interest are the degrees of mental well-being and the enhancement or alteration of the sexual self-concept.", does not provide the specificity required in a SR and should be divided. Authors should consider the degrees of mental well-being as a primary outcome, citing validated scales that need to have been used in RCTs to be included in the SR (Ex: The Warwick-Edinburgh Mental Well-being Scale (WEMWBS), Ryff Psychological Well -being (PWBS-42) scale). In this way, they would consider enhancement and alteration of the sexual self-concept as secondary outcomes, also including validated scales that should be present in RCTs, such as The Multidimensional Sexual Self-Concept Questionnaire (MSSCQ).

The inclusion in the methodology of the PICOS strategy (P= Adults, I= pole dancing, C= usual activity, or other type of physical activity/dance, O= mental well-being, sexual self-concept assessed by the scales, S= RCT) will further enrich this section.

7. PLOS authors have the option to publish the peer review history of their article (what does this mean?). If published, this will include your full peer review and any attached files.

Reviewer #1: **Yes: **Ricardo Ney Cobucci

---

## [Author Response · Author response to Decision Letter 0]

16 Jun 2024

Dear Reviewer #1:

Thank you very much for your careful review! We have carefully studied your valuable comments and have made amendments one by one. We hope that these adjustments will meet your requirements!

1.“The introduction is long and tiring. The suggestion for the authors is to summarize the content featuring the main theme, pole dancing, followed by text with the most up-to-date references of studies that show potential psychological benefits, maintaining the important gap of there being no systematic reviews specifically on pole dancing and concluding the introduction with the main objective.”

We have streamlined the introduction. We cut out a paragraph. The paragraph reads as follows:

“Additionally, dance is critical in augmenting bodily awareness, fostering mental health, and facilitating neurorehabilitation(7). The release of endorphins triggered by dance activities has been linked to significant improvements in mental well-being(8). In essence, dance offers a holistic therapeutic avenue for enhancing mental health, embodying various psychological benefits such as improved self-esteem, stress mitigation, emotional health, and heightened bodily consciousness. This extensive body of research underscores dance's therapeutic potential in advancing mental health and supporting psychological well-being across various populations. ”

In addition, we have removed the following from the introduction:

“The multifaceted nature of dance, characterized by its rhythm, musicality, social engagement, technical and physical demands, and capacity to facilitate connection, mindfulness, and the expression of aesthetic emotions, contributes to its overall health and well-being benefits.(9)”

“The intersectionality of pole dancing with feminist theory further highlights its role in dialogues concerning gender, sexuality, and empowerment(13). ”

2.Finally, the methodology is well constructed, but the text "Our primary outcomes of interest are the degrees of mental well-being and the enhancement or alteration of the sexual self-concept.", does not provide the specificity required in a SR and should be divided. Authors should consider the degrees of mental well-being as a primary outcome, citing validated scales that need to have been used in RCTs to be included in the SR (Ex: The Warwick-Edinburgh Mental Well-being Scale (WEMWBS), Ryff Psychological Well -being (PWBS-42) scale). In this way, they would consider enhancement and alteration of the sexual self-concept as secondary outcomes, also including validated scales that should be present in RCTs, such as The Multidimensional Sexual Self-Concept Questionnaire (MSSCQ).

We adjusted part of the article and identified mental wellbeing and sexual self-concept as the primary outcome and secondary outcome respectively. Finally, we have included in the article specific scales used in the measurement process. For details, please see lines 134-140.

3.The inclusion in the methodology of the PICOS strategy (P = Adults, I = pole dancing, C = usual activity, or other type of physical activity/dance, O = mental well-being, sexual self-concept assessed by the scales, S = RCT) will further enrich this section.

We have adopted the "PICOS" principle to present the inclusion criteria according to your suggestion.For details, please see lines 126-143. 

Thank you again for your valuable advice. We have further refined our protocol and made the upcoming studies more standardized.

---

## [Decision Letter · Decision Letter 1]

2 Jul 2024

Impact of pole dancing on mental wellbeing and sexual self-concept: protocol for a systematic review and meta-analysis

PONE-D-24-13949R1

Dear Dr. Li,

We’re pleased to inform you that your manuscript has been judged scientifically suitable for publication and will be formally accepted for publication once it meets all outstanding technical requirements.

Kind regards,

Adetayo Olorunlana, Ph.D.

Academic Editor

PLOS ONE

Additional Editor Comments (optional):

Reviewers' comments:

Reviewer's Responses to Questions

**Comments to the Author**

1. Does the manuscript provide a valid rationale for the proposed study, with clearly identified and justified research questions?

Reviewer #1: Yes

2. Is the protocol technically sound and planned in a manner that will lead to a meaningful outcome and allow testing the stated hypotheses?

Reviewer #1: Yes

3. Is the methodology feasible and described in sufficient detail to allow the work to be replicable?

Reviewer #1: Yes

4. Have the authors described where all data underlying the findings will be made available when the study is complete?

Reviewer #1: Yes

5. Is the manuscript presented in an intelligible fashion and written in standard English?

Reviewer #1: Yes

6. Review Comments to the Author

You may also provide optional suggestions and comments to authors that they might find helpful in planning their study.

Reviewer #1: The authors responded to the reviewers' suggestions and the manuscript is ready to be published. Congratulations!

7. PLOS authors have the option to publish the peer review history of their article (what does this mean?). If published, this will include your full peer review and any attached files.

Reviewer #1: **Yes: **Ricardo Ney Cobucci

---

## [Editor Report · Acceptance letter]

5 Jul 2024

PONE-D-24-13949R1 

PLOS ONE

Dear Dr. Li, 

I'm pleased to inform you that your manuscript has been deemed suitable for publication in PLOS ONE. Congratulations! Your manuscript is now being handed over to our production team.

Kind regards, 

on behalf of

Associate Professor Adetayo Olorunlana 

Academic Editor

PLOS ONE